# *Botrytis cinerea* PMT4 Is Involved in *O*-Glycosylation, Cell Wall Organization, Membrane Integrity, and Virulence

**DOI:** 10.3390/jof11010071

**Published:** 2025-01-17

**Authors:** Verónica Plaza, Alice Pasten, Luz A. López-Ramírez, Héctor M. Mora-Montes, Julia Rubio-Astudillo, Evelyn Silva-Moreno, Luis Castillo

**Affiliations:** 1Laboratorio de Biología Molecular y Bioquímica, Departamento de Biología, Universidad de La Serena, La Serena 1700000, Chile; vplaza@userena.cl (V.P.); alice.pasten@userena.cl (A.P.); 2Departamento de Biología, División de Ciencias Naturales y Exactas, Universidad de Guanajuato, Guanajuato 36050, Mexico; adrianalr@ugto.mx (L.A.L.-R.); hmora@ugto.mx (H.M.M.-M.); 3Instituto de Ciencias Biomédicas, Universidad Autónoma de Chile, Santiago 7500912, Chile; julia.rubio@uautonoma.cl; 4GEMA Center for Genomics, Ecology and Environment, Faculty of Interdisciplinary Studies, Universidad Mayor, Santiago 7510041, Chile; evelyn.silva@umayor.cl

**Keywords:** pathogenicity, antifungal drugs, cell membrane, glycoprotein

## Abstract

Proteins found within the fungal cell wall usually contain both *N*- and *O*-oligosaccharides. *N*-glycosylation is the process where these oligosaccharides (hereinafter: glycans) are attached to asparagine residues, while in *O*-glycosylation the glycans are covalently bound to serine or threonine residues. The *PMT* family is grouped into *PMT1*, *PMT2*, and *PMT4* subfamilies. Using bioinformatics analysis within the *Botrytis cinerea* genome database, an ortholog to *Saccharomyces cerevisiae* Pmt4 and other fungal species was identified. The aim of this study was to assess the relevance of the *bcpmt4* gene in *B. cinerea* glycosylation. For this purpose, the *bcpmt4* gene was disrupted by homologous recombination in the B05.10 strain using a hygromycin B resistance cassette. Expression of *bcpmt4* in *S. cerevisiae* Δ*Scpmt4* or Δ*Scpmt3* null mutants restored glycan levels like those observed in the parental strain. The phenotypic analysis showed that Δ*bcpmt4* null mutants exhibited significant changes in hyphal cell wall composition, including reduced mannan levels and increased amounts of chitin and glucan. Furthermore, the loss of *bcpmt4* led to decreased glycosylation of glycoproteins in the *B. cinerea* cell wall. The null mutant lacking *PMT4* was hypersensitive to a range of cell wall perturbing agents, antifungal drugs, and high hydrostatic pressure. Thus, in addition to their role in glycosylation, the *PMT4* is required to virulence, biofilm formation, and membrane integrity. This study adds to our knowledge of the role of the *B. cinerea bcpmt4* gene, which is involved in glycosylation and cell biology, cell wall formation, and antifungal response.

## 1. Introduction

*Botrytis cinerea* is an anamorphic, phytopathogenic fungus responsible for causing gray mold disease on a wide variety of dicotyledonous plants, including fruits, vegetables, and ornamental species [1,2,3,4]. Fungicides with different modes of action are currently used to control *B. cinerea* in agricultural industries, such as Phenylpyrroles (Fludioxonil), Hydroxyanilides (Fenhexamid), Dicarboxamides (Iprodione), Succinate Dehydrogenase inhibitors (Boscalid), DeMethylation inhibitors (Tebuconazole), and methionine biosynthesis inhibitors (Pyrimethanil) [4]. On the other hand, the food industry employs heat or high hydrostatic pressure (HHP) to control filamentous fungi like *B. cinerea* to extend the shelf life of food products [5,6,7].

The fungal cell wall (FCW) is a highly dynamic structure that protects against physical and chemical stressors and plays crucial roles during infection. In many species, the FCW is composed of an inner skeletal layer made up of various polysaccharides, including β-glucans (e.g., β-1,3-, mixed β-1,3-/β-1,4-, β-1,6-glucans), α-glucans, galactomannans, and chitin, which is surrounded by a layer that is densely packed with glycoproteins that are also vital to fungi [8,9,10,11,12,13,14].

The glycoproteins are attached to the wall by covalent and non-covalent linkages. The outer layer is removed using detergents or chaotropic agents, while the inner layer is extracted by degrading the polysaccharides with hydrolytic enzymes. The two types of covalently linked proteins that have been found are glycosyl phosphatidylinositol (GPI-dependent wall proteins) and PIR proteins (proteins with internal repeats) [14,15,16,17]. Glycoproteins usually contain both *N*- and *O*-linked oligosaccharides. The *N*-linked glycans are attached to asparagine residue, while *O*-linked glycans are attached to serine or threonine protein residue. However, the number of genes involved in both glycosylation processes appears to be higher in yeast than in filamentous fungi. The genome of *B. cinerea* contains more than 100 different genes encoding putative glycoproteins that can be anchored to the cell wall through GPI anchor to β1,3-glucan or β1,6-glucan or the plasma membrane, in line with other fungal species where a similar gene number has been found [18,19]. PIR proteins are highly *O*-glycosylated and are attached directly to β1,3-glucan via an ester linkage to a deamidated glutamine residue in the repeat sequence, possible through a transglutaminase-type of reaction [20,21]. In silico analysis of the *B. cinerea* genome showed that two PIR proteins contain the conserved PIR-repeat sequence (Q[IV]XDGQ[IVP]Q) [19].

This analysis also found conserved family genes involved in *N*- and *O*-glycosylation. The protein-*O*-mannosyltransferases are involved in coupling of the first mannose residue to the hydroxyl side groups of serine and threonine amino acids, involving two conserved gene families: the *PMT* and *KRE2/MNT1* families. The *PMT* family is further divided into three subfamilies, namely *PMT1*, *PMT2*, and *PMT4*, with the number of members in each subfamily differing across species. Only three *PMT* genes are found in *B. cinerea*, one in each of the subgroups [19,22,23,24]. The lower number of *PMT* genes in *Botrytis* is similar to other filamentous ascomycetes, such as *Aspergillus nidulans*, *Neurospora crassa*, and *Fusarium gramineum* [23,25,26]. In some fungi, *PMT2* is essential for growth [27], and crucial for cell wall integrity [26,28]. Both *PMT1* and *PMT4* are induced during cell wall regeneration [29,30], and are required for normal cell wall composition and virulence [22,31,32,33]. In filamentous fungi, deletion of individual *PMTs* has been reported. Deletions of *Trichoderma reesei pmtI*, *A. fumigatus pmt1*, *A. fumigatus pmt2*, *A. nidulans pmtA*, and *A. awamori pmtA* were not lethal but affected hyphal growth, cell wall integrity, and development [23,24,34,35,36,37]. Evidence for the relevance of *O*-glycosylation in *B. cinerea* has come from a study on *pmt* genes, which encode proteins that function in the *O*-linked glycosylation pathway and were shown to be required for fungal pathogenicity [38]. This evidence suggests that Pmt4 plays a pivotal role in fungal pathogenesis, cell wall integrity, and plant penetration [22,31,38]. Nevertheless, certain aspects of cell wall structure and its role in conferring resistance to chemical antifungals or maintaining membrane integrity in *B. cinerea* have yet to be fully elucidated.

In this study, we describe the importance of the cell surface *O*-glycosylation and its role in the *B. cinerea* cell wall. To establish the role of the *B. cinerea bcpmt4* gene in the *O*-glycosylation of this fungus, we characterized single null mutants and performed functional complementation analyses of this gene in glycosylation mutants of *Saccharomyces cerevisiae*.

## 2. Materials and Methods

### 2.1. B. cinerea and S. cerevisiae Strains, Growth Conditions, and Genomic DNA Isolation

Strain B05.10 of *B. cinerea* was used as a wild-type (WT) strain and recipient for genetic modifications [39,40]. The *S. cerevisiae* strains that were used in this study (relevant genotypes are indicated) are as follows: BY4742 (WT), YOR321W (Δ*pmt3*), YOR321W- pVTU260 (Δ*pmt3*), YOR321W-pVTU260-*bcpmt4* (Δ*pmt3/bcpmt4*), YJR143C (Δ*pmt4*), YJR143C-pVTU260 (Δ*pmt4*), and YJR143C-pVTU260-*bcpmt4* (Δ*pmt4*/*bcpmt4*). The mutant strains are isogenic to and derived from WT strain BY4742 (MATα *his3*Δ*1 leu2*Δ*0 lys2*Δ*0 ura3*Δ*0*). The *S. cerevisiae* strains were obtained from the EUROSCARF gene deletion collection (http://www.euroscarf.de/index.php?name=Description, accessed on 12 December 2015). The fungal strains were kept at −80 °C in a preservation tube containing 25% glycerol.

*B. cinerea* strains were grown on malt agar (MEA) (Merck) or in synthetic minimal medium (MM) modified by Cotoras, Folch [41]. Conidia production was measured from cultures on MEA, and each strain was seeded with 10 μL of conidia suspensions (2.5 × 10^5^ conidia/mL); the isolate was incubated in MEA plates for 2 weeks at 20 °C, under 24 h photoperiod (12 h light/12 h darkness). A conidial suspension of each strain was prepared in sterile water and filtered through a sterile 5 mL pipette tip containing glass wool. The conidia were collected and filtered through Miracloth (Merck, Rahway, NJ, USA), resuspended in 10 mL sterile water, and counted with a Neubauer camera [42].

*S. cerevisiae* strains were grown in yeast extract peptone dextrose (YPD), extract peptone galactose YPG, or yeast nitrogen (YNB) base without amino acid (Difco, Beirut, Lebanon) and supplements as needed. Cultures were grown at 30 °C for 24 h [42].

For DNA isolation, *B. cinerea* mycelium was harvested, submerged into liquid nitrogen, and ground into a powder. Genomic DNA extraction was performed using standard protocols [42]. DNA pellets were dissolved in 50 μL of TE (10 mM Tris-HCl [pH 8.0], 1 mM EDTA) and quantified using a Nanodrop spectrophotometer (Thermo Fisher Scientific, Carlsbad, CA, USA).

### 2.2. Generation of a B. cinerea Δpmt4 Strain

Plasmid pLOB7 was generously provided by Dr. Jan van Kan (Wageningen University, The Netherlands). The plasmid pUC18-hph was constructed according to [42]. The selection cassette was amplified by PCR using primers hph-fw and hph-rv (Appendix A) and cloned into the *BamH*I and *Sal*I restriction sites of plasmid pUC18 to generate pUC18-hph. The primers used in this study are listed in Appendix A. The disruption cassette pUC18-hph-*pmt4* was generated by fusing two 0.5 kb PCR-generated fragments, carrying part of the *PMT4* ORF and part of its terminator, respectively, to hph in the pUC18-hph plasmid. The entire replacement cassette was amplified using primers P1fw and P4rv and used to transform *B. cinerea* strain B05.10.

Protoplast generation and transformation were carried out as previously described [43]. The homologous integration cassette was verified by PCR. To check the integration, the flanking region upstream of *bcpmt4* and part of the *hph* marker gene were amplified using primers P1fw and P2rv. The 3’ fusion point was checked by amplifying a fragment using primers P3fw and P4rv. Homologous integration into the correct locus occurred in two out of the three transformants analyzed (Appendix A). Determination of the copy number of the hygromicin B resistance gene was carried out as previously described by Aguayo, Riquelme [44], with a single confirmed integration event at the target locus (Appendix A).

### 2.3. Real-Time PCR

The qPCR amplification and analysis were performed using a Light Cycler 96 (Roche, Diagnostics, Indianapolis, IN, USA) instrument, 2× KAPA SYBR kit (Sigma-Aldrich, Burlington, MA, USA), and the CT was determined by relative method, in a total volume of 20 µL, following the manufacturer’s recommendations. The following two-step qPCR conditions were employed: activation at 95 °C for 2 min; 40 cycles of denaturation for 5 s at 95 °C; and annealing and elongation for 30 s at 60 °C. Melting curve analysis was performed at the end of the amplification. Serial dilutions of both plasmids were used to generate a standard curve from 2.0 × 10^1^ to 2.0 × 10^9^ copies/µL. Each standard dilution was analyzed in triplicate, followed by plotting of CT values, thus generating a standard curve by linear regression. PCR amplification efficiency (E) was calculated according to Rasmussen [45] and compared using Lightcycler 96 software data (Appendix A) [46].

### 2.4. Construction of bcpmt4 Complementation Strain

A complementation strategy following [42,47] was used. Briefly, a DNA fragment containing *bcpmt4*, including the 1500 bp upstream and 300 bp downstream of the coding region, was amplified using primers Pattb1 and Pattb2. The purified DNA fragment was recombined with the pNR4 plasmid (generously provided by Dr. Jan van Kan) in BP reactions (Invitrogen, Waltham, MA, USA) in the appropriate concentration [47]. The resulting plasmid was used for *B. cinerea* protoplast transformation. Homologous integration of the *pmt4* locus and its expression were verified by PCR and RT-PCR (Appendix A).

### 2.5. Construction of bcpmt4 Expression Plasmid for S. cerevisiae Complementation Assays

To obtain the coding region of *bcpmt4* without intron sequences, the total RNA from *B. cinerea* strain B05.10 was isolated using a total RNA kit (Omega, USA, Norwalk, CT, USA). cDNA was generated by reverse transcription–polymerase chain reaction (RT-PCR), using a SuperScript One-Step RT-PCR System (Invitrogen, Waltham, MA, USA). The *bcpmt4* coding region was amplified from the cDNA by PCR using Expand High Fidelity Taq and Pfu polymerases (Roche, Indianapolis, IN, USA) and primers Pex-fw and P4rv. The *bcpmt4* gene was cloned in the *NcoI* and *BamHI* restriction sites of plasmid pVTU260 (*URA3* auxotrophic marker (EUROCARF, Stockholm, Sweden)), which contains the yeast *ADH1* promoter for gene expression, yielding plasmid pVTU260-*bcpmt4*.

The pVTU260-*bcpmt4* plasmid, as well as the empty vector, were transformed into *S. cerevisiae* Δ*Scpmt3* or Δ*Scpmt4* null mutant, respectively, following [48]. Yeast transformants were selected on YNB without amino acid containing 2% glucose without uracil, and were used for phenotypic cell wall-related assays, following [48].

### 2.6. Quantification of Cell Wall Carbohydrates

The strains were cultivated at 20 °C in 2% [wt/vol] Malt Extract Broth (MEB) medium for 24 h with shaking at 200 rpm. The cells were disrupted using glass beads in a FastPrep machine (MP Biomedicals, Santa Ana, CA, USA), and the resulting homogenate was centrifuged at 8500× *g* for 10 min to isolate the pellet containing cell debris and walls. This pellet was washed with 1 M NaCl, resuspended in a buffer containing 500 mM Tris-HCl (pH 7.5), 2% [wt/vol] SDS, 0.3 M β-mercaptoethanol, and 1 mM EDTA, then boiled for 10 min and freeze-dried. For glucose and mannose quantification, the cell walls were hydrolyzed in 2 M trifluoroacetic acid and boiled for 3 h. Chitin content was determined by hydrolyzing the cell walls in 6N HCl at 100 °C for 17 h [49]. Quantification of sugar monomers from acid-hydrolyzed walls was performed using an HPAEC-PAD with a Dionex Bio-LC system [49].

### 2.7. Sensitivity to Cell Wall Perturbing Agents

The sensitivity of *B. cinerea* to cell wall perturbing agents was performed according to [42]. Conidia were collected and washed with sterile water in a concentration of 2.5 × 10^5^ conidia/mL. Aliquots of 100 μL of 2.5 × 10^5^ conidia/mL were added to sterile polystyrene 96-well microtiter plates (JetBiofil, Guangzhou, China) with MEB containing Congo red (CR, 500 μg/mL), Calcofluor white (CFW, 500 μg/mL), or caffeine (500 μg/mL). The plates were incubated at 20 °C for 72 h, and the extent of growth was quantified at 590 nm (Victor X3 Perkin Elmer 2030 workstation, Perkin Elmer, Shelton, CT, USA). The sensitivity of cells to drugs were confirmed by testing the effects of a serial dilution on plate agar. Ten-fold serial dilutions were prepared and 3 μL of each dilution was spotted onto MEA (control) and MEA containing CR (900 μg/mL), CFW (800 μg/mL) or caffeine (1000 μg/mL). All experiments were performed in triplicate.

The sensitivity to Zymolyase was determined using a modified version of a previously described assay [42,50].

The sensitivity of *S. cerevisiae* strains towards the addition of CFW or CR in the growth medium was tested in a spot assay similar to that described for *B. cinerea* [42]. Cells of each strain were resuspended at OD_600_ = 1.0, and a series of ten-fold dilutions were prepared and spotted on YPD agar containing either CFW (200 μg/mL) or CR (200 μg/mL) [51]. Petri dishes were incubated for 24 h at 28 °C.

To evaluate the antifungal susceptibility, we tested three chemical groups of antifungals, namely Boscalid, Iprodione, and Fenhexamid, as previously described by [4]. Aliquots containing 100 μL of 2.5 × 10^5^ conidia/mL were inoculated in plastic 96 wells with MEB, Boscalid (19.2 mg/L, 9.6 mg/L, 4.8 mg/L, 2.4 mg/L, 1.2 mg/L, and 0.6 mg/L), Iprodione (7.2 mg/L, 3.6 mg/L, 1.8 mg/L, 0.9 mg/L, 0.45 mg/L, and 0.225 mg/L), and Fenexamid (23.04 mg/L 11.5 mg/L, 5.76 mg/L, 2.88 mg/L, 1.44 mg/L and 0.72 mg/L). The plates were incubated at 20 °C for 72 h and quantified at 590 nm (Victor X3 Perkin Elmer 2030 workstation).

### 2.8. Plant Infection Tests

Leaf infection tests were performed on tomato and apple fruits following Doehlemann, Berndt [52]. Fruit tissues were punctured with a 21G syringe, sterilized in 75% ethanol, and inoculated with 5 µL of 2.5 × 10⁵ conidia/mL suspension. After 4 days at 20 °C, lesion sizes were measured and internal damage was assessed by halving the apple fruits.

### 2.9. Membrane Integrity Assay

A suspension of 5 × 10^5^ onidia/mL was collected by centrifuging at 8500× *g* for 10 min at 25 °C, then washed twice with 50 mM sodium phosphate buffer (pH 7.0) and centrifuged at 8500× *g* for 2 min. It was stained with 10 µg/mL propidium iodide (PI) for 5 min at 30 °C, washed again to remove excess dye, and observed under a light microscope with an epifluorescence system (Eclipse E-200, Nikon, Nishioi, Shinagawa-ku, Tokyo) [53].

### 2.10. Biofilm Formation and Quantification

Aliquots of 100 µL of 2.5 × 10⁵ conidia/mL were incubated in 96-well plates at 20 °C for 72 h. After washing off unbound cells, 0.1% methyl violet was added for 5 min to stain the adhered biofilm. The wells were washed with PBS and measured at 590 nm, (Victor X3 PerkinElmer 2030 workstation, Perkin Elmer, Shelton, CT, USA) with OD values reflecting the biofilm’s hyphae and extracellular polymeric content [42,53].

### 2.11. High Hydrostatic Pressure (HHP) Assay

High hydrostatic pressure was performed following [54]. A concentration of 1 × 10^4^ conidia/mL of B05.10 or *bcpmt4* mutants or reintegrant *B. cinerea* were used in this experiment. The samples were packed in flexible polyethylene bags and processed in HHP equipment (capacity 2 L cylindrical container, Avure Technologies Incorporated, Kent, WA, USA). Pressure level and pressurization time were controlled automatically, and the samples were pressurized at 150, 250, and 400 MPa for 2 and 4 min, respectively, according to each trial, at room temperature (20 ± 2 °C). The control sample was an unpressurized sample at atmospheric pressure (0.1 MPa) and room temperature.

Microbiological analysis of different samples treated with HHP was performed following [55]. To mold enumeration 1.0 mL of the initial (10^−1^) dilution was spread-plated on three different plates of Dichloran Rose Bengal Chloramphenicol (DRBC) agar (Difco, Detroit, MI, USA), and 0.1 mL of each subsequent dilution was spread on individual DRBC plates. The inoculated plates were then incubated at 20 °C for 5 days, and plates with 15–300 colonies were counted.

### 2.12. Preparation of Wall and Soluble Fractions

The purified cell wall was prepared according to the procedure described by [20]. Fungal cells were collected and washed twice with distilled water, suspended in 1 mM phenylmethylsulphonyl fluoride (PMSF), and broken with glass beads in a FastPrep-24 instrument (MP Biomedicals, Santa Ana, CA, USA) (5 mg beads/mg cells). The cell walls were washed four times with chilled distilled water, then boiled for 10 min (twice) with 2% SDS in distilled water, and finally washed six times more with chilled 1.0 mM PMSF in distilled water. Isolated walls were freeze-dried (frozen at −20 °C, vacuum pressure 0.120 mbar and condensing temperature −50 °C) using FreeZone 2.5 Liter Benchtop Freeze Dryers (Labconco, London, UK), and stored at −20 °C until further use.

The cell walls were suspended in 2.5 mL of a solution of 0.01 M ammonium acetate, pH 6.3, containing 2% *v*/*v* of β-2-mercaptoethanol, shaken for 3 h at 28 °C, and then pelleted. The supernatant was concentrated by freeze-drying and the total protein contents of the samples were determined by the Bradford method using bovine serum albumin as standard.

### 2.13. Gel Electrophoresis and Glycoprotein Stain

The proteins were separated by SDS–PAGE onto a precast, gradient, 4–20% polyacrylamide gel using Tris-Glycine-SDS running buffer. The same gel was processed with the Pro-Q Emerald 300 Glycoprotein stain kit (Molecular Probes, Invitrogen, Waltham, MA, USA), as described by [56]. The Glycoprotein bands were visualized under a UV light system (modelo U1000 Labnet, Edison, NJ, USA). To verify equal sample loading, the same gel was subsequently stained with Coomassie Blue (Thermo Fisher Scientific, Waltham, MA, USA).

### 2.14. Statistical Analysis

All the experiments were repeated at least three times. One-way analysis of variance (ANOVA) (Statgraphics Plus^®^ 5.1 software, Statistical Graphics Corp., Herndon, VA, USA) was used to demonstrate significant differences among samples. Significance testing was performed using Fisher’s least significant difference (LSD) test and Mann–Whitney U test differences were accepted with *p*-values < 0.05 [53].

## 3. Results

### 3.1. The B. cinerea PMT4

The 2307 bp *bcpmt4* (769 aa) open reading frame (GenBankTM/EBI accession ID XP_024546940) was predicted before as orthologs of *ScPMT4* [19,22]. In addition, Pmt4 is predicted to be an integral membrane protein with multiple transmembrane domains, the hydropathy profiles with those of *S. cerevisiae PMT*s is shown in Figure 1 [57,58].

### 3.2. The bcpmt4 Complements a S. cerevisiae Δpmt4 Null Mutant

We carried out genetic complementation to find out whether the function of *bcpmt4* is similar to that of *S. cerevisiae PMT4*. The 2307 bp cDNA fragment, containing the entire *bcpmt4* coding sequence was cloned into the yeast expression vector pVTU260. The resulting construction and the empty vector were used to transform the *S. cerevisiae* Δ*pmt3* and Δ*pmt4* mutants, which are defective for *O*-glycosylation [28]. *S. cerevisiae* Δ*pmt4* mutant is sensitive to CFW and CR, but the expression of *B. cinerea bcpmt4* in this mutant background restored the phenotype to WT levels (Figure 2A).

To confirm these results, cell wall analysis in *S. cerevisiae* BY4742, *pmt3*Δ, *pmt4*Δ, and complemented strains (*Scpmt4*Δ + *bcpmt4, Scpmt3*Δ + *bcpmt4*) were carried out. In the β-ME cell wall extracts, high molecular weight highly polydispersed material of 150–200 kDa was evident in BY4742 strain and complement strains, together a 40 kDa band, except in the null mutants *Scpmt3*Δ and *Scpmt4*Δ, indicating a significant decrease in glycosylation of cell wall glycoproteins, but the expression of *bcpmt4* in these mutants restores the WT phenotype (Figure 2B). Our findings suggest that *B. cinerea bcpmt4* is a functional ortholog of *S. cerevisiae PMT3* and *PMT4*.

### 3.3. Deletion of B. cinerea bcpmt4 Led to Alterations in Hyphal Wall Composition and Glycosylation Deficiencies

To ascertain whether *O*-glycosylation plays a role in the *B. cinerea* cell wall, a simple deletion mutant of putative *bcpmt4* genes was obtained. The strategy used was a targeted mutagenesis using the gene replacement method described by Kars, McCalman [43] and B05.10 as WT strain. Later, the protoplasts transformed with a gene replacement construct, containing a hygromycin B selection marker cassette flanked by *bcpmt4* sequences was obtained, and a single conidia isolation was used to confirm the targeted deletions (Appendix A). The analyses performed were those described by Aguayo, Riquelme [44], with a single confirmed integration event at the target locus (Appendix A).

The deletion of *bcpmt4* from the haploid B05.10 strain showed a significant alteration in its hyphal wall composition. A reduction of 88% in mannan levels in Δ*bcpmt4* mutant, respectively. Also, chitin and glucan levels increased in the mutant cells (Table 1). Our data are consistent with other studies, showing a drop in mannan and an increase in wall chitin and glucan content [59]. To determine whether mannan changes may be related to the cell wall glycosylation, the cell walls of *B. cinerea* strains were treated with hot SDS, and then with β-2-mercaptoethanol. The released glycoproteins were separated by SDS–PAGE and detected with Pro-Q Emerald 300 Glycoprotein stain. Figure 3 shows that the mutant showed reduced glycan levels in the glycoprotein when compared to the B05.10 or complemented strains, indicating that the *bcpmt4* gene could be involved in *O*-glycosylation on *B. cinerea*.

### 3.4. Cell Wall Defects of the Δbcpmt4 Mutant

It has been demonstrated that *S. cerevisiae* strains with defects in cell wall biosynthesis are hypersensitive to Calcofluor White (a reagent known to bind to β-1,4-linked polysaccharides, specifically chitin and cellulose), Congo red (a compound that binds to glucan, interfering with proper cell wall construction), caffeine, and Zymolyase (which affects cell wall integrity as a result of the presence of a main β-1,3-glucanase activity and a residual protease activity). These agents interfere with cell wall construction and stress response mechanisms [26,27,60,61,62].

To verify the cell wall defect in the *B. cinerea* Δ*pmt*4 mutants, the WT and mutants were grown on solid and liquid media containing CFW, CR, or caffein. As shown in Figure 4A, the Δ*bcpmt4* mutant was sensitive to caffeine when compared to the B05.10 or complemented strains (Figure 4A). In addition, upon treatment with Zymolyase, only 54% Δ*bcpmt4* conidia survived after 7 h of treatment, whereas the survival of the WT control strain or complemented strains was more than 80% (Figure 4B). These results demonstrate that the Δ*bcpmt4* mutant is hypersensitive to Zymolyase, indicating a defect in its cell wall structure. These results show that deletion of Δ*bcpmt4* in *B. cinerea* causes cell wall defects.

### 3.5. Membrane Integrity Defects in the Δbcpmt4 Mutant

To characterize the Δ*bcpmt4* mutant phenotypes in more detail, we studied the effect of glycosylation reduction on membrane integrity loss in *B. cinerea,* by fluorescent staining with propidium iodide (PI) and High hydrostatic pressure (HHP) stress response. Cellular membrane disruption allows PI to enter the cell, where it binds to DNA, and consequently, cells are fluorescent under fluorescence microscopy. According to the data in Figure 5A, the plasmatic membrane of conidia of *B. cinerea* Δ*bcpmt4* mutant was markedly damaged, in comparison to the parental strain B05.10 or complemented strains in minimal medium, where the integrity of the membrane declined after 3 h and 4 h of incubation.

HHP exerts effects upon yeast or bacterial cells, interfering with cell membranes or cellular architecture [63,64]. To determine whether glycosylation is involved in membrane stability during HHP stress, the parental and Δ*bcpmt4* mutant strains were subjected to high-pressure treatments at 100 and 200 MPa. Figure 5B shows the immediate effect of both treatments. From 100 MPa or 200 MPa, lower growth of *B. cinerea* Δ*bcpmt4* mutant was observed, compared to the control or complemented strain. These data show that glycosylation is involved in cell membrane stability on HHP stress.

### 3.6. Deletion of bcpmt4 in B. cinerea Leads to Hypersensitivity to Antifungal Drugs

Fungicides with various modes of action are currently employed to control *B. cinerea*; however, no information has been associated with genes involved in glycosylation. To assess whether the cell wall defect in the *B. cinerea* Δ*pmt4* mutants can change the sensitivity to these fungicides, the strains were incubated with different concentrations of Hydroxyanilides (Fenhexamid), Dicarboxamides (Iprodione), and Succinate Dehydrogenase inhibitors (Boscalid). The results show that the *B. cinerea* Δ*pmt4* mutant was highly sensitivity to Iprodione from 2 up to 7 mg/L, compared to the parental strain or complemented strains. A similar result was found in Fenhexamid up to 20 mg/L. However, for Boscalid, a significative difference was only observed at 20 mg/L (Figure 6). These results show a possible correlation between cell wall integrity and hypersensitivity to these antifungal drugs.

### 3.7. Lack of bcpmt4 Affects Conidia and Biofilm Formation Capacity

For the assessment of conidia production, Δ*bcpmt4* mutant and WT *Botrytis* strains were incubated in MAE at 20 °C under a regular photoperiod for 24 h (12 h light/12 h darkness) to induce conidiation. As shown in Figure 7A,C, the mutant produced viable conidia; but the conidia number was reduced on average by 62% when compared to the WT strain or complemented strains. Complementary to this, the conidia produced by the Δ*bcpmt4* mutant did not show aggregation (Figure 7B), suggesting that cell surface alterations are not involved in this process.

Regarding biofilm formation, this is considered an important virulence factor for pathogenic fungi. It has been described that yeasts and filamentous fungi can stick to biotic and abiotic surfaces, and the morphological patterns can be affected by environmental parameters. According to the studies carried out in *C. albicans* mutants, glycosylation is essential for proper biofilm formation [65]. Therefore, we analyzed the biofilm-forming capacity of the mutant generated in this study. The WT and Δ*bcpmt4* mutant mycelia were growing and analyzed in polystyrene microtiter plates. In the Δ*bcpmt4* mutant strain, we observed a 66% reduction in biofilm formation when compared to WT or complemented strains (Figure 7D,E). These observations suggested that the deletion of *bcpmt4* caused alteration in conidia and biofilm formation capacity.

### 3.8. O-Glycosylation Is Required for Virulence on Fruit

The glycoproteins in the cell wall play a key role in the host–fungus interaction and the outcome during the infection in *C. albicans*, *U. maydis*, *B. cinerea*, *Candida glabrata*, *C. parapsilosis,* and *A. fumigatus* [10,22,31,34,38,59,66,67]. To confirm the results obtained by Gonzalez, Brito [38], indicating that *bcpmt4* is involved in the pathogenicity of *B. cinerea,* tomato and apple fruits were infected by inoculating 5 μL of 2.5 × 10^5^ conidia/mL of Δ*bcpmt4* and WT strains. Four days post-inoculation (dpi), disease development was measured as the diameter of the expanding lesions. The deletion of *bcpmt4* displayed a statistically significant decrease in radial lesion sizes in comparison with WT in tomato and apple fruits (Figure 8A,B). Based on the findings obtained in this research and the obtained by Gonzalez, Brito [38], it can be suggested that *O*-glycosylation contributes significantly to the virulence of *B. cinerea*.

## 4. Discussion

In *S. cerevisiae* or *C. albicans*, the loss of genes involved in glycosylation results in hypersensitivity to the cell wall-perturbing agents CW and CR [28,68]. The phenotypic severity of the mutant suggests that missing carbohydrate chains are required for normal cell growth [69]. The glycoprotein profile in the Δ*Scpmt4* and Δ*Scpmt3* mutants released by β-ME reflects changes in glycan content, where a missing protein with molecular weight around 40 kDa was observed, described previously by [20] PIR4. Further, the expression of the *B. cinerea bcpmt4* gene into Δ*Scpmt4* or Δ*Scpmt3* mutant restores glycan levels in the glycoprotein, with the presence of PIR4, suggesting that this enzyme is involved in *O*-glycosylation in *B. cinerea*.

The loss of *PMTs* in *S. cerevisiae*, *C. albicans*, *A. fumigatus,* and *Ustilago maydis* cause several defects, such as abnormal morphology and hyphal growth, and cellular aggregation [26,27,61]. The *B. cinerea* Δ*bcpmt4* null mutant has characteristic phenotypes associated with disruption of protein mannosylation, including defects in the cell wall composition with reduction in mannan and an increase in the amount of chitin [42,66,70,71,72,73]. The reduction in mannan was confirmed in the glycoprotein releases by β-2-mercaptoethanol in cell wall mutants. An increase in the amount of chitin in the cell wall is often observed in mutants that are more sensitive to CFW and CR, which is viewed as part of a mechanism induced by fungi to compensate for the loss of cell wall integrity [49,59,66,73,74,75]. The increased susceptibility to the cell-wall perturbing agent observed here could be an effect of cell wall damage in the mutant. This has been confirmed by an increased zymolyase sensitivity—a typical response of mutants in *S. cerevisiae* defect in glucan component—or by a decreased amount of mannoprotein [76,77,78,79]. Similar results were obtained by Gonzalez, Brito [38], using different *bcpmt* mutants in *B. cinerea*. However, cell wall alteration may also explain the reduction in the adhesion on polystyrene surfaces by *B. cinerea* Δ*bcpmt4* null mutants [80,81]. Doss, Potter [82,83] described that *B. cinerea* conidia adhere to the host and can secrete a sheath fungal and adhesive material composed, in part, of glucose, protein, and polysaccharide. This mucilage could allow *Botrytis* to adhere to the tomato cuticle immediately upon conidia hydration. Equally important, a growing number of fungal cell wall proteins have been demonstrated to act as cell surface adhesins and confer binding to diverse ligands, and the majority of fungal adhesins are glycoproteins [80,84,85]. Therefore, glycosylation in *B. cinerea* seems to be required for fungal binding to polystyrene surfaces or on plant surfaces, thereby playing a pivotal role in pathogenesis [38].

Most glycoproteins are plasma membrane-associated cell wall and secreted proteins and several environmental stresses, such as temperature, ethanol, oxidative, and osmotic stress, as well as HHP, cause alterations in the physical properties of the membrane lipids in the cells [86,87,88]. In *S. cerevisiae*, HHP induces change in the cell membrane and the gene *HSP12* is induced when the cells are exposed to 200 MPa, this gene has been associated with cell wall plasticizer and membrane destabilization [63]. The inactivation of yeast and filamentous fungi like *B. cinerea* at higher pressures (generally > 250 to 600 MPa) is required, wherein the ascospores are generally inactivated. However, it appears that reduced glycosylation in the *B. cinerea* mutant was made more susceptible to HHP at 100 or 200 MPa, which could be associated with the alteration of the membrane permeability in this mutant. This is the first time that *O*-glycosylation is related to HHP stress response in *B. cinerea*.

Our data suggest that the increased hypersensitivity to Iprodione or Fenhexamid observed in the Δ*bcpmt4* null mutant could be an associated with cell membrane or cell wall damage in the mutant. It is well known that the Iprodione damages the integrity of the mycelial cell wall and cell membrane in *Bipolaris maydis* by inhibiting enzymes involved in cell wall production [89]. On the other hand, Fenhexamid inhibits C3-ketoreductase, which is involved in ergosterol biosynthesis [90]. This cell wall or membrane alteration may also explain the reduced adhesion or HHP stress response in the mutant.

The survival of fungal species partly depends on their ability to produce large quantities of viable conidia, a process regulated by genetic, nutritional, and environmental factors. Recent studies have also shown that *N*-glycosylated proteins play a role during the vegetative growth stage, being involved in nutrient utilization, cell wall biogenesis, and redox processes. Moreover, at the conidium formation stage, the *N*-glycosylated proteins were involved in glycogen storage, cell wall biogenesis, and at the appressorium formation stage [91], which is consistent with our data showing that deletion of *bcpmt4* in *B. cinerea* caused reduction conidia production. Attenuated virulence has been documented in mutants defective in *N*- and *O*-linked glycosylation; coincidently, some mutants have also been affected in adhesion [68,72,81,92]. Furthermore, our results have shown that the *B. cinerea* Δ*bcpmt4* mutant had significantly decreased virulence in tomato and apple fruits, with similar observations made by Gonzalez, Brito [38]. This phenotype could be associated with low adhesion and cell wall alteration in the mutants. Defects in glycosylation may compromise protein functions relevant to host–pathogen interactions and virulence [80,85,91].

## 5. Conclusions

In summary, the *B. cinerea bcpmt4* is the functional ortholog of *S. cerevisiae PMT4*. In addition, the loss of *bcpmt4* results in reduced glycosylation of the glycoprotein present in the *B. cinerea* cell wall, as well as cell wall alterations, which represent a reduction in adhesion and membrane integrity. These phenotypes could be responsible for the reduction in conidia production, hypersensitivity to HHP or synthetic fungicides, and pathogenicity in the *B. cinerea* mutant. These studies expand the current knowledge about the role of the *B. cinerea bcpmt4* in *O*-glycosylation, cell biology, cell wall formation, and antifungal response.

## Figures and Tables

**Figure 1 jof-11-00071-f001:**
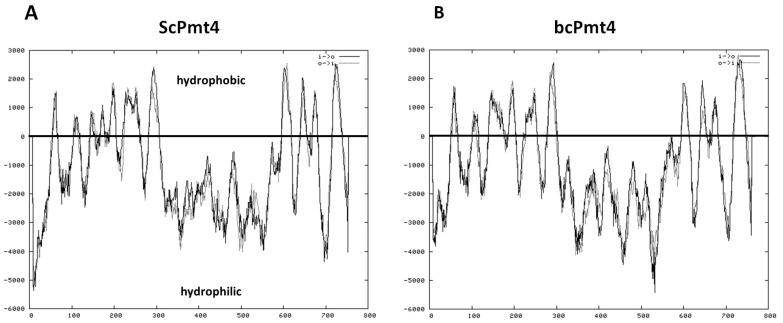
Scheme of transmembrane domains predicted in *S. cerevisiae* and *B. cinerea Pmt*4. The predictions of transmembrane segments were performed using the TMPRED programs. (**A**,**B**) represent the hydropathy profiles of *ScPmt4* and *bcPmt4*, respectively. Hydropathy analysis of *B. cinerea* Pmt4 showed a similar profile compared to the *S. cerevisiae* protein.

**Figure 2 jof-11-00071-f002:**
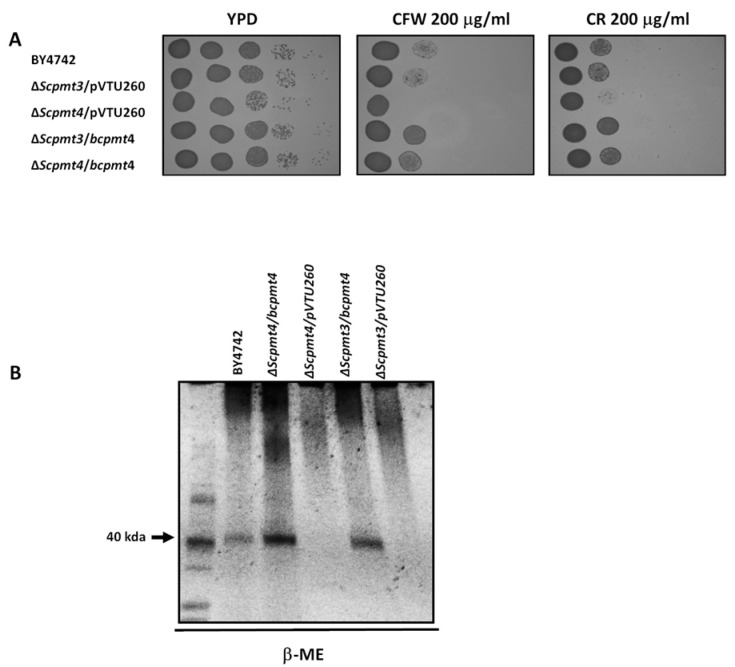
*B. cinerea bcpmt4* encodes a functional ortholog of *S. cerevisiae PMT*4. (**A**) Spot assays showing sensitivities of a WT *S. cerevisiae* strain (BY4742), Δ*pmt3* and Δ*pmt4* mutants transformed with empty pVTU260 plasmid (Y0R321W/pVTU260 and YJR143C/pVTU260), Δ*pmt3* and Δ*pmt4* mutants transformed with pVTU260/*bcpmt4* cDNA (Y0R321W/*bcpmt4* and YJR143C/*bcpmt4*). Control (YPD without drugs), CFW 200 μg/mL (YPD with 200 μg/mL of Calcofluor white), and CR 200 μg/mL (YPD with 200 μg/mL of Congo red). (**B**) Analysis by SDS–PAGE of the materials extracted by β-2-mercaptoethanol (β-ME) from the cell walls of *S. cerevisiae pmt3*Δ and *pmt4*Δ mutants and complemented strains. Gel stained with Pro-Q Emerald 300 Glycoprotein stain.

**Figure 3 jof-11-00071-f003:**
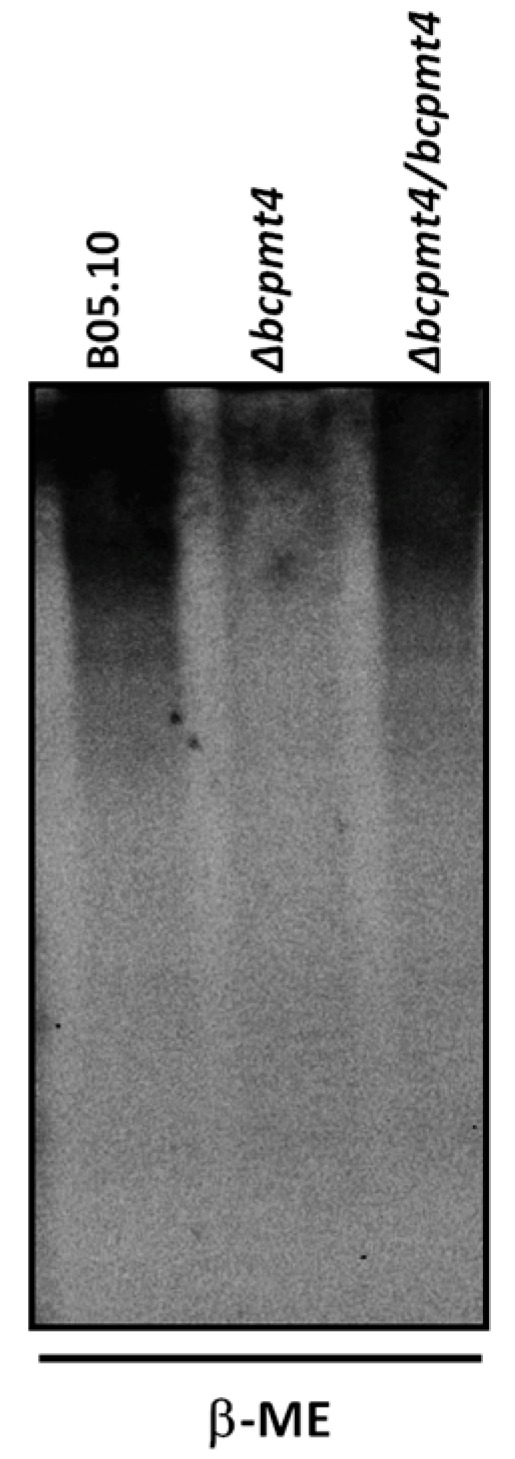
Analysis by SDS–PAGE of the glycoproteins present in *B. cinerea* strains. Glycoproteins extracted by β-2-mercaptoethanol from the cell walls of *B. cinerea* B05.10, Δ*bcpmt4* and complemented strain were electrophoresed at a constant current. Gel stained with Pro-Q Emerald 300 Glycoprotein stain.

**Figure 4 jof-11-00071-f004:**
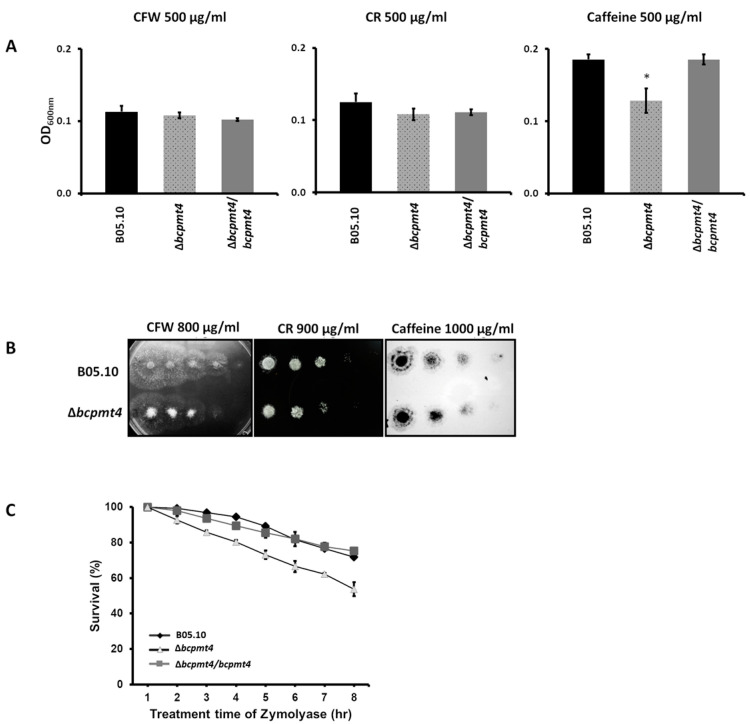
The *B. cinerea O*-glycosylation mutant is hypersensitive to cell wall perturbing agents and Zymolyase. (**A**) Sensitivity to CFW, CR, or caffeine of the B05.10 WT, Δ*bcpmt4* mutant, and the complemented strains (Δ*bcpmt4* + *bcpmt4*). Cells were incubated in malt extract in the presence or absence of 500 µg/mL CFW, CR, or caffeine for 72 h at 20 °C. (**B**) Sensitivity to CFW, CR, or caffeine of the B05.10 WT strain and mutants on agar plates. (**C**) Zymolyase sensitivity. Conidia (2.5 × 10^5^ conidia/mL) of the B05.10 WT strain, mutants, and the complemented strain were incubated with 200 μg/mL Zymolyase 20T at 25 °C, and the decrease in OD_590_ nm was monitored over time (represented as conidia surviving the treatment; survival %). The data represents the mean ± SD from three independent experiments. The asterisk indicates significant differences (*p* < 0.05, Mann–Whitney U test) compared with the parental strain (* WT vs. Δbcpmt4).

**Figure 5 jof-11-00071-f005:**
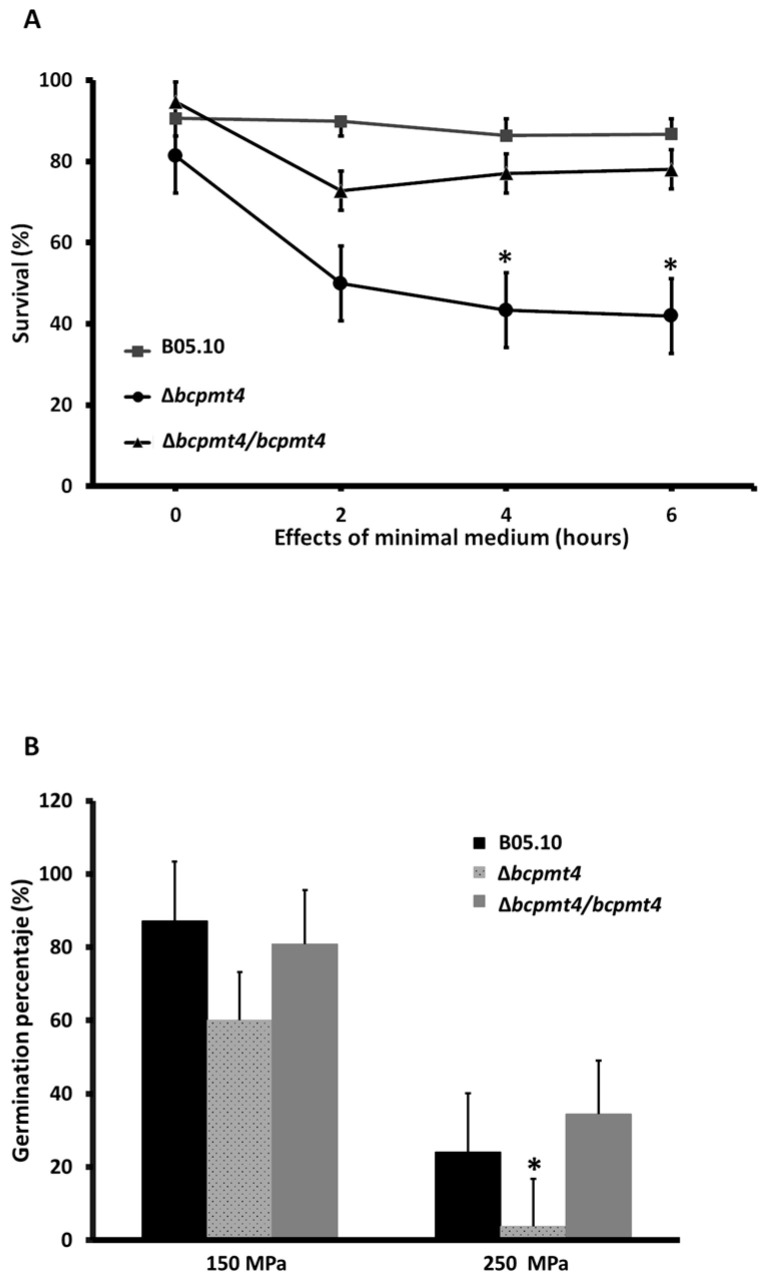
Glycosylation is important for membrane integrity. (**A**) Effects of the minimal medium on plasma membrane integrity of conidia of *B. cinerea* strains. (**B**) Survival percentage of *B. cinerea* after treatment of high hydrostatic pressure (150 MPa/4 min, 250 MPa/4 min) in solid media incubation for 48 h at 20 °C. The asterisk indicates significant differences (*p* < 0.05, Mann–Whitney U test) compared with the parental strain (* WT vs. Δbcpmt4).

**Figure 6 jof-11-00071-f006:**
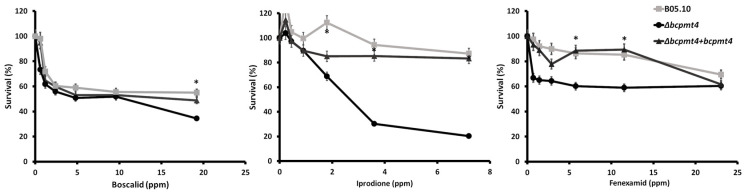
The *bcpmt4* mutants display sensitivity to synthetic fungicides. The sensitivity of the B05.10 WT, Δ*bcpmt4*, and complemented strains was determined quantitatively by the broth dilution method. The agents to which the Δ*bcpmt4* mutant demonstrated hypersensitivity are shown (Boscalid, Iprodione, and Fenexamid). The asterisk indicates significant differences (*p* < 0.05, Mann–Whitney U test) compared with the parental strain (* WT vs. Δbcpmt4).

**Figure 7 jof-11-00071-f007:**
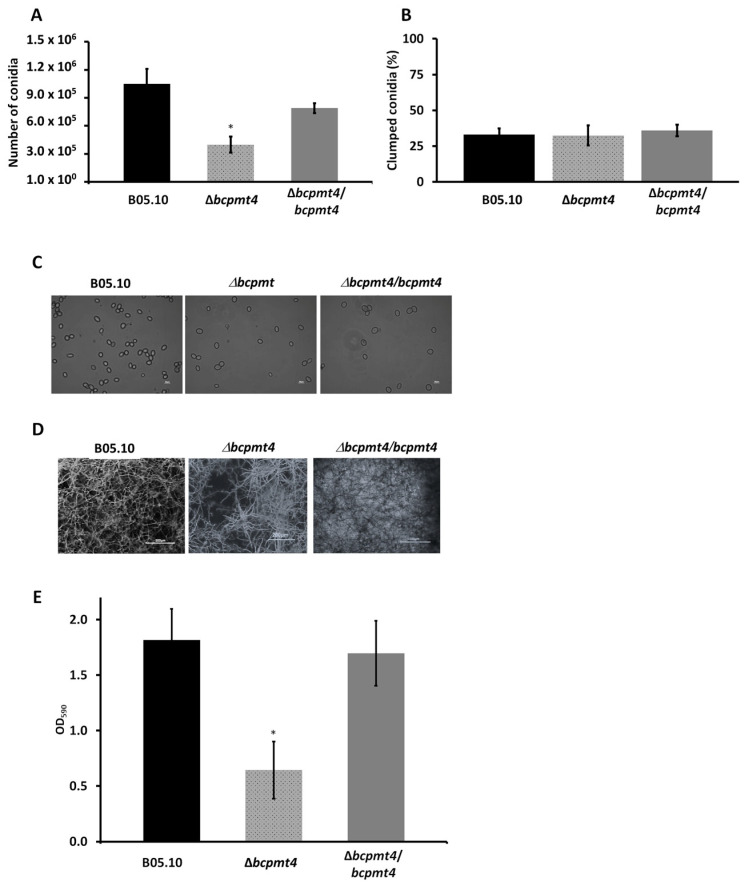
Deletion of *bcpmt4* affects conidia and biofilm formation. (**A**) Quantification of conidia production in the B05.10 WT, Δ*bcpmt4*, and complemented strains grown for two weeks on MEA at 20 °C under a 24 h photoperiod (12 h light/12 h darkness). (**B**) Quantification of conidial clumping. Percentages of conidia present in groups of two or more clumped conidia together were counted. Bars represent the mean ± SD. (**C**) Light microscopic (40×) examination of conidial suspensions indicated that Δ*bcpmt4* conidia did not exhibit clumping (the white line represent 50 mm). (**D**) Biofilm formation of the B05.10 WT, Δ*bcpmt4*, and complemented strains on polystyrene after incubation for 72 h at 20 °C. Bound mycelia were stained with Crystal violet and visualized by fluorescence microscopy. (**E**) Quantification of biofilm formation by measuring OD_590_. Bars represent the mean ± SD of three independent experiments measured in triplicate. The asterisk indicates significant differences (*p* < 0.05, Mann–Whitney U test) compared with the parental strain (* WT vs. Δbcpmt4).

**Figure 8 jof-11-00071-f008:**
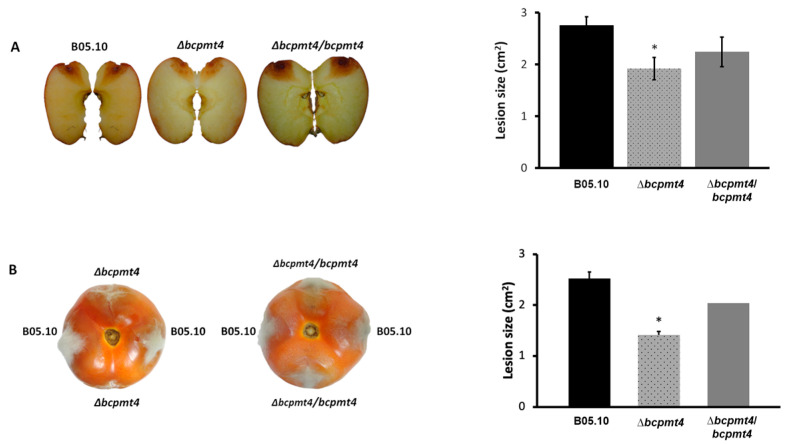
Virulence of the *B. cinerea* Δ*bcpmt4* in comparison to the wild-type B05.10. (**A**,**B**) The tissues (apple or tomato) were wounded with a pinprick, followed by the application of 5 µL droplets (containing 2.5 × 10⁵ conidia/mL) for inoculation. Lesion sizes were measured at 4 days post-inoculation (dpi) at 20 °C. Bars represent the mean lesion diameter ± SD. The asterisk indicates significant differences (*p* < 0.05, Mann–Whitney U test) compared to the parental strain (* WT vs. Δbcpmt4).

**Table 1 jof-11-00071-t001:** Chemical analysis of mutant cell wall.

Strain	Glucose	Mannose	Glucosamine
B05.10	0.469 ± 0.03 (46.9 ± 3%) *	0.340 ± 0.01 (34.0 ± 1%) *	0.007 ± 0.00 (7.0%) *
Δ*bcpmt4*	0.771 ± 0.04 (77.1 ± 4%) *	0.039 ± 0.06 (39.0 ± 6%) *	0.035 ± 0.02 (3.5%) *

The values are measured in mg/mg of cell wall dry weight ± S.D. (n = 3). * Data are shown as percentage of cell wall dry weight.

## Data Availability

The original contributions presented in the study are included in the article/Appendix A, further inquiries can be directed to the corresponding author.

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
