# Peer review of "Botrytis cinerea PMT4 Is Involved in O-Glycosylation, Cell Wall Organization, Membrane Integrity, and Virulence"

_jof, 2025, doi:10.3390/jof11010071_

Round 1
Reviewer 1 Report
There is a meaningful research. The bcpmt4 gene was disrupted by homologous recombination in the B05.10 strain using a hygromycin B resistance cassette. The expression of bcpmt4 in S. cerevisiae Scpmt4Δ or Scpmt3Δ null mutants restored glycan levels similar to that observed in the parental strain. The phenotypic analysis showed that bcpmt4Δ null mutants showed a significant alteration in hyphal cell wall composition, including reduced mannan levels and an increase in the amount of chitin and glucan. But minor modifications are needed.
1.Abstract is needed to be modified. The structure is objective, method, result, and conclusion. The objective is long but unclear. For the objective, we need to explain the purpose and task of the research, so that readers can understand the starting point and destination of the research. Method, describe the theories, ideas, approaches, and methods adopted by the research. Result, overview of the main findings.
2.Line 200-202: In pesticide concentrations, ppm (parts per million) is no longer used, please change mg/L or ml/L.
3.The fig.1, 3, 7 were not clear, please modify.
4. The format of References needs to be standardized. Line 795 “Strahl-Bolsinger, S. and A. Scheinost”, line 804 “Proszynski, T.J., K. Simons, and M. Bagnat” , line 806, “Ram, A.F., et al.,”…
5.Fig4, To verify the cell wall defect in the B. cinerea pmt4â–² mutants, the WT, and mutants were grown on solid and liquid media containing CFW, CR, or Caffein. The bcpmt4â–² mutant was sensitive to CR and caffeine when compared to the B05.10 or complemented strains. The CR 500 µg/ml result is not significant so is the conclusion right?
Author Response
There is a meaningful research. The bcpmt4 gene was disrupted by homologous recombination in the B05.10 strain using a hygromycin B resistance cassette. The expression of bcpmt4 in S. cerevisiae Scpmt4Δ or Scpmt3Δ null mutants restored glycan levels similar to that observed in the parental strain. The phenotypic analysis showed that bcpmt4Δ null mutants showed a significant alteration in hyphal cell wall composition, including reduced mannan levels and an increase in the amount of chitin and glucan. But minor modifications are needed.
1.Abstract is needed to be modified. The structure is objective, method, result, and conclusion. The objective is long but unclear. For the objective, we need to explain the purpose and task of the research, so that readers can understand the starting point and destination of the research. Method, describe the theories, ideas, approaches, and methods adopted by the research. Result, overview of the main findings.
Answer Reviewer: We thank the reviewer for their comment and have modified the abstract.
2.Line 200-202: In pesticide concentrations, ppm (parts per million) is no longer used, please change mg/L or ml/L.
Answer Reviewer: We modified the units from ppm to mg/mL as suggested (see line 214-216).
- The fig.1, 3, 7 were not clear, please modify.
Answer Reviewer: Figure legends were modified to enhance comprehension (line 313, 392 and 586).
- The format of References needs to be standardized. Line 795 “Strahl-Bolsinger, S. and A. Scheinost”, line 804 “Proszynski, T.J., K. Simons, and M. Bagnat” , line 806, “Ram, A.F., et al.,”…
Answer Reviewer: The references were standardized
5.Fig4, To verify the cell wall defect in the B. cinerea pmt4â–² mutants, the WT, and mutants were grown on solid and liquid media containing CFW, CR, or Caffein. The bcpmt4â–² mutant was sensitive to CR and caffeine when compared to the B05.10 or complemented strains. The CR 500 µg/ml result is not significant so is the conclusion right? (V Plaza, solo CR es significativo)
Answer Reviewer: The text was revised to avoid confusion, and only the caffeine effect was statistically significant (line 408).

Reviewer 2 Report
The work provides valuable insights into the involvement of the PMT4 protein in Botrytis cinerea. Numerous bioassays were conducted to confirm the activity . Despite the manuscript's excellent quality, some revisions and additions are needed.
- Correct the issues highlighted in the manuscript text.
- Provide detailed information about the primers used.
- Adjust the placement of figures throughout the text to ensure proper alignment.
- Separate the results from the discussion; in some sections, the results appear to be part of the discussion.
- Review and revise the figure legends for clarity and completeness.
- In Supplementary Fig. S1: Please specify the mutants being referred to.
- In Table S1: Indicate the origin of the primers used.
- In Table S2: Clarify that the "Estimated hygromycin copy number (HCN)" was determined by relative quantification qPCR in bioassays with...

Author Response
The work provides valuable insights into the involvement of the PMT4 protein in Botrytis cinerea. Numerous bioassays were conducted to confirm the activity . Despite the manuscript's excellent quality, some revisions and additions are needed.
- Correct the issues highlighted in the manuscript text.
Answer Reviewer: We thank the reviewer for their comment. The highlighted text has been modified.
The B. cinerea bcpmt4 gene orthologs to the ScPMT4 gene in S. cerevisiae was identified and characterized.
The expression of bcpmt4 gene into Scpmt4Δ or Scpmt3Δ mutants restore glycan level in the glycoprotein, suggesting that this enzyme is involved in O-glycosylation.
The bcpmt4Δ null mutants exhibited significant changes in hyphal cell wall composition, including reduced mannan levels and increased amounts of chitin and glucan.
The bcpmt4Δ null led to decreased glycosylation of glycoproteins in the B. cinerea cell wall
The null mutant lacking PMT4 was hypersensitive to a range of cell wall perturbing agents, antifungal drugs and high hydrostatic pressure.
The PMT4 is required to virulence, biofilm formation and membrane integrity
This research enhances our understanding of the B. cinerea bcpmt4 gene's role in glycosylation, cell biology, cell wall formation, and antifungal responses.
- Provide detailed information about the primers used.
Answer Reviewer: References have been incorporated into Table S1.
- Adjust the placement of figures throughout the text to ensure proper alignment.
Answer Reviewer: The figures were alignment.
- Separate the results from the discussion; in some sections, the results appear to be part of the discussion.
Answer Reviewer: The paragraph has been moved to the Discussion section.
- Review and revise the figure legends for clarity and completeness.
Answer Reviewer: Figure legends were revised to enhance clarity and comprehension.
- In Supplementary Fig. S1: Please specify the mutants being referred to.
Answer Reviewer: The Fig S1 legend was clarified.
- In Table S1: Indicate the origin of the primers used.
Answer Reviewer: References have been incorporated into Table S1.
- In Table S2: Clarify that the "Estimated hygromycin copy number (HCN)" was determined by relative quantification qPCR in bioassays with...
Answer Reviewer: Additional information has been incorporated into Table S2.

Reviewer 3 Report
In this study, to establish the role of the B. cinerea pmt4 gene in the O-glycosylation of this fungus, the authors characterized single null mutants and performed functional complementation analyses of this gene in glycosylation mutants of S. cerevisiae and described the importance of the cell surface O-glycosylation and its role in B. cinerea cell wall. The results of this study contained some novel and interesting data, and they were good and adequate enough to support the conclusion. In general, the manuscript is well organized. But the authors should pay more attention to the abbreviation and type errors throughout the manuscript. Before accepting for publication, however, the following suggestions need to be addressed.
General comments
1 In the “Abstract”, supplement the full name of B. cinerea.
2 Gene name should use italic, such as “bcpmt4” in Line 25, “bcpmt4Δ mutant” in Line 32 and “Trichoderma reesei pmtI, Aspergillus. fumigatus pmt1, A. fumigatus pmt2, A. nidulans pmtA” in Line 88.
3 In the “Introduction”, the authors introduce that PMT2 is essential for growth, and crucial for cell wall integrity. Both PMT1 and PMT4 are induced during cell wall regeneration, and are required for normal cell wall composition and virulence. However, it does not explain why this study selected PMT4 as the research object, and it is necessary to add more introduction and research progress on the function of PMT4.
4 In the “Materials and Methods”, the experimental section lacks sufficient detail. To ensure that your experiments can be fully reproduced, please consider expanding this section with more comprehensive descriptions of the methodologies used, such as 2.6, 2.8, 2.9 and 2.10.
5 There should be a space between the number and the unit, such as “20℃” in lines 111,118 146 147,184…, please check and modify other similar problems in the manuscript.
6 The nomenclature of the mutant is generally 'Δ' add the gene name. In the manuscript, some places are written 'Δpmt4', such as '2.2. Generation of a B. cinerea Δpmt4 strain' and 'Δbcpmt4' in Fig. 8B; in other parts of the manuscript, it is 'bcpmt4Δ', and it is recommended to uniformly modify it to the format of 'Δbcpmt4'.
7 In the “Results”, for Fig. 8B, the difference in lesion size between WT and mutant strains after inoculation was not clearly seen in the map. It is recommended that WT, mutant and complementary strains be inoculated on different fruits and photographed.
8 In the “Discussion”, line 605, please discuss the contents of the references in detail, rather than simply saying 'following [38]'.
Specific comments
1. Line 25, “S. cerevisiae” should be “S. cerevisiae”.
2. Line 88, “Aspergillus. fumigatus” should be “Aspergillus fumigatus”.
3. Line 116, “S. cerevisiae strains” should be “S. cerevisiae strains”.
4. In line 160, “RT-PCR” should be “reverse transcription-polymerase chain reaction (RT-PCR)”, because this is the first time in the manuscript; in line 164, “reverse transcription-polymerase chain reaction (RT-PCR)” should be “RT-PCR”.
5. Please check the writing of “℃” in the manuscript, such as lines 207, 446, 459, 533….
6. In lines 261, 285, “Fig 1” should be “Fig.1”, in line 307, “Fig 2” should be “Fig.2”, please check and modify similar problems in the manuscript.
7. In line 332, "WT" is better than " wild-type (WT)" because you have already used the abbreviation in the text.
8. In lines 335-336, “by Aguayo et al. (2011)”, supplement the number of the reference.
9. In line 447, “OD590” should be “OD590”.
10. In line 544, supplement the full name of C. glabrata.
11. In “p < 0.05”, “p” should be italic, please check and modify the manuscript.
In this study, to establish the role of the B. cinerea pmt4 gene in the O-glycosylation of this fungus, the authors characterized single null mutants and performed functional complementation analyses of this gene in glycosylation mutants of S. cerevisiae and described the importance of the cell surface O-glycosylation and its role in B. cinerea cell wall. The results of this study contained some novel and interesting data, and they were good and adequate enough to support the conclusion. In general, the manuscript is well organized. But the authors should pay more attention to the abbreviation and type errors throughout the manuscript. Before accepting for publication, however, the following suggestions need to be addressed.
General comments
1 In the “Abstract”, supplement the full name of B. cinerea.
2 Gene name should use italic, such as “bcpmt4” in Line 25, “bcpmt4Δ mutant” in Line 32 and “Trichoderma reesei pmtI, Aspergillus. fumigatus pmt1, A. fumigatus pmt2, A. nidulans pmtA” in Line 88.
3 In the “Introduction”, the authors introduce that PMT2 is essential for growth, and crucial for cell wall integrity. Both PMT1 and PMT4 are induced during cell wall regeneration, and are required for normal cell wall composition and virulence. However, it does not explain why this study selected PMT4 as the research object, and it is necessary to add more introduction and research progress on the function of PMT4.
4 In the “Materials and Methods”, the experimental section lacks sufficient detail. To ensure that your experiments can be fully reproduced, please consider expanding this section with more comprehensive descriptions of the methodologies used, such as 2.6, 2.8, 2.9 and 2.10.
5 There should be a space between the number and the unit, such as “20℃” in lines 111,118 146 147,184…, please check and modify other similar problems in the manuscript.
6 The nomenclature of the mutant is generally 'Δ' add the gene name. In the manuscript, some places are written 'Δpmt4', such as '2.2. Generation of a B. cinerea Δpmt4 strain' and 'Δbcpmt4' in Fig. 8B; in other parts of the manuscript, it is 'bcpmt4Δ', and it is recommended to uniformly modify it to the format of 'Δbcpmt4'.
7 In the “Results”, for Fig. 8B, the difference in lesion size between WT and mutant strains after inoculation was not clearly seen in the map. It is recommended that WT, mutant and complementary strains be inoculated on different fruits and photographed.
8 In the “Discussion”, line 605, please discuss the contents of the references in detail, rather than simply saying 'following [38]'.
Specific comments
1. Line 25, “S. cerevisiae” should be “S. cerevisiae”.
2. Line 88, “Aspergillus. fumigatus” should be “Aspergillus fumigatus”.
3. Line 116, “S. cerevisiae strains” should be “S. cerevisiae strains”.
4. In line 160, “RT-PCR” should be “reverse transcription-polymerase chain reaction (RT-PCR)”, because this is the first time in the manuscript; in line 164, “reverse transcription-polymerase chain reaction (RT-PCR)” should be “RT-PCR”.
5. Please check the writing of “℃” in the manuscript, such as lines 207, 446, 459, 533….
6. In lines 261, 285, “Fig 1” should be “Fig.1”, in line 307, “Fig 2” should be “Fig.2”, please check and modify similar problems in the manuscript.
7. In line 332, "WT" is better than " wild-type (WT)" because you have already used the abbreviation in the text.
8. In lines 335-336, “by Aguayo et al. (2011)”, supplement the number of the reference.
9. In line 447, “OD590” should be “OD590”.
10. In line 544, supplement the full name of C. glabrata.
11. In “p < 0.05”, “p” should be italic, please check and modify the manuscript.
Author Response
In this study, to establish the role of the B. cinerea pmt4 gene in the O-glycosylation of this fungus, the authors characterized single null mutants and performed functional complementation analyses of this gene in glycosylation mutants of S. cerevisiae and described the importance of the cell surface O-glycosylation and its role in B. cinerea cell wall. The results of this study contained some novel and interesting data, and they were good and adequate enough to support the conclusion. In general, the manuscript is well organized. But the authors should pay more attention to the abbreviation and type errors throughout the manuscript. Before accepting for publication, however, the following suggestions need to be addressed.
General comments
1 In the “Abstract”, supplement the full name of B. cinerea.
Answer Reviewer: We thank the reviewer for their comment. We added the full name to the abstract (Line 21).
2 Gene name should use italic, such as “bcpmt4” in Line 25, “bcpmt4Δ mutant” in Line 32 and “Trichoderma reesei pmtI, Aspergillus. fumigatus pmt1, A. fumigatus pmt2, A. nidulans pmtA” in Line 88.
Answer Reviewer: Fungal names have been italicized (Line 88).
3 In the “Introduction”, the authors introduce that PMT2 is essential for growth, and crucial for cell wall integrity. Both PMT1 and PMT4 are induced during cell wall regeneration, and are required for normal cell wall composition and virulence. However, it does not explain why this study selected PMT4 as the research object, and it is necessary to add more introduction and research progress on the function of PMT4.
Answer Reviewer: Additional information has been incorporated in the introduction (Line 92-96).
4 In the “Materials and Methods”, the experimental section lacks sufficient detail. To ensure that your experiments can be fully reproduced, please consider expanding this section with more comprehensive descriptions of the methodologies used, such as 2.6, 2.8, 2.9 and 2.10.
Answer Reviewer: Additional information has been incorporated into sections 2.6, 2.8, 2.9, and 2.10.
5 There should be a space between the number and the unit, such as “20℃” in lines 111,118 146 147,184…, please check and modify other similar problems in the manuscript.
Answer Reviewer: A space has been inserted between the number and the unit.
6 The nomenclature of the mutant is generally 'Δ' add the gene name. In the manuscript, some places are written 'Δpmt4', such as '2.2. Generation of a B. cinerea Δpmt4 strain' and 'Δbcpmt4' in Fig. 8B; in other parts of the manuscript, it is 'bcpmt4Δ', and it is recommended to uniformly modify it to the format of 'Δbcpmt4'.
Answer Reviewer: The mutant nomenclature was changed to Δbcpmt4 in the manuscript.
7 In the “Results”, for Fig. 8B, the difference in lesion size between WT and mutant strains after inoculation was not clearly seen in the map. It is recommended that WT, mutant and complementary strains be inoculated on different fruits and photographed.
Answer Reviewer: To eliminate potential behavioral differences attributable to the fruit, the wild-type strain B05.10 was infected with the mutant or the reintegrant in the same fruit. To enhance observation, additional light was added to the tomato fruit photograph.
8 In the “Discussion”, line 605, please discuss the contents of the references in detail, rather than simply saying 'following [38]'.
Answer Reviewer: The paragraph has been modified (Line 664).
Specific comments
- Line 25, “S. cerevisiae” should be “S. cerevisiae”.
Answer Reviewer: The suggested change has been incorporated into the manuscript.
- Line 88, “Aspergillus. fumigatus” should be “Aspergillus fumigatus”.
Answer Reviewer: The dot has been eliminated from the manuscript (Line 88)
- Line 116, “S. cerevisiae strains” should be “S. cerevisiae strains”
Answer Reviewer: The italics were removed from 'strains” (Line 120).
- In line 160, “RT-PCR” should be “reverse transcription-polymerase chain reaction (RT-PCR)”, because this is the first time in the manuscript; in line 164, “reverse transcription-polymerase chain reaction (RT-PCR)” should be “RT-PCR”.
Answer Reviewer: The reviewer's suggestion has been incorporated into the text.
- Please check the writing of “℃” in the manuscript, such as lines 207, 446, 459, 533….
Answer Reviewer: We have incorporated the degree Celsius symbol ('℃') into the manuscript.
- In lines 261, 285, “Fig 1” should be “Fig.1”, in line 307, “Fig 2” should be “Fig.2”, please check and modify similar problems in the manuscript.
Answer Reviewer: The suggested change has been incorporated into the manuscript.
- In line 332, "WT" is better than " wild-type (WT)" because you have already used the abbreviation in the text.
Answer Reviewer: The suggested change has been incorporated into the manuscript (Line 359).
- In lines 335-336, “by Aguayo et al. (2011)”, supplement the number of the reference.
Answer Reviewer: The reference number has been incorporated (Line 362-363).
- In line 447, “OD590” should be “OD590”.
Answer Reviewer: The suggested change has been incorporated into the manuscript (Line 471).
- In line 544, supplement the full name of C. glabrata.
Answer Reviewer: Full name has been incorporated into the manuscript (Line 602).
- In “p < 0.05”, “p” should be italic, please check and modify the manuscript.
Answer Reviewer: The suggested change has been incorporated into the manuscript.

Round 2
Reviewer 1 Report
There has been a significant improvement in revised Ms, but I think it is needed minor revision:
1. As for genes, such as PMT(s), please keep italics and consistent in lowercases or capital letters in the manuscript.
2. There should be spaces between numbers and units (Except ℃). Please read the full text carefully and make the necessary changes.
3. Please read the full text and modify some minor issues, such as 1) Line 224, A a suspension of …; 2) Line 181 and 226 8 500 xg(There is space between 8 and 5), Line 202 Caffeine (1000 mg /ml, no space between 1 and 0) and Line 159 1,500 bp (a , between 1 and 5); 3) ml (Line 202, mg /ml) or mL(line 197, mg/mL), Please make sure.
1. As for genes, such as PMT(s), please keep italics and consistent in lowercases or capital letters in the manuscript.
Such as: capital letters Line 30, PMT4, Line 81, 82, 84 to 86, 292, 334……
lowercase letters: Line 32, B. cinerea bcpmt4, Line88, 98, 139……
capital and lowercase letters: Line 87 Pmts, Line 314, 316(Pmt4 )……
2. There should be spaces between numbers and units (Except ℃). Please read the full text carefully and make the necessary changes.
Space (Right): Line 114,117, 119 (numbers) mL (numbers) mL…
No Space (need revise): Line 125 10mM Tris-HCl, 126 1mM EDTA…
No Space (Right): Line 115 20°C, 210 at 28°C..
Space (need revise): Line 242 (20 ± 2 °C), Line 248 20 °C…….
3. Please read the full text and modify some minor issues, such as 1) Line 224, A a suspension of …; 2) Line 181 and 226 8 500 xg(There is space between 8 and 5), Line 202 Caffeine (1000 mg /ml, no space between 1 and 0) and Line 159 1,500 bp (a , between 1 and 5); 3) ml (Line 202, mg /ml) or mL(line 197, mg/mL), Please make sure.
Author Response
Major comments
There has been a significant improvement in revised Ms, but I think it is needed minor revision:
- As for genes, such as PMT(s), please keep italics and consistent in lowercases or capital letters in the manuscript.
A: Thank you for the reviewer's feedback. We have ensured that the manuscript maintains a consistent format.
- There should be spaces between numbers and units (Except ℃). Please read the full text carefully and make the necessary changes.
A: The manuscript has been reviewed, and we have made the necessary changes to include a space between numbers and units.
- Please read the full text and modify some minor issues, such as 1) Line 224, A a suspension of …; 2) Line 181 and 226 8 500 xg(There is space between 8 and 5), Line 202 Caffeine (1000 mg /ml, no space between 1 and 0) and Line 159 1,500 bp (a , between 1 and 5); 3) ml (Line 202, mg /ml) or mL(line 197, mg/mL), Please make sure.
A: We thank the reviewer for their comment. We have addressed the minor issues raised and resolved them accordingly (Line 160, 182, 198, 203, 225).
- As for genes, such as PMT(s), please keep italics and consistent in lowercases or capital letters in the manuscript.
Such as: capital letters Line 30, PMT4, Line 81, 82, 84 to 86, 292, 334……
lowercase letters: Line 32, B. cinerea bcpmt4, Line88, 98, 139……
capital and lowercase letters: Line 87 Pmts, Line 314, 316(Pmt4 )……
A: We have addressed the changes as suggested by the reviewer.
- There should be spaces between numbers and units (Except ℃). Please read the full text carefully and make the necessary changes.
Space (Right): Line 114,117, 119 (numbers) mL (numbers) mL…
No Space (need revise): Line 125 10mM Tris-HCl, 126 1mM EDTA…
No Space (Right): Line 115 20°C, 210 at 28°C..
Space (need revise): Line 242 (20 ± 2 °C), Line 248 20 °C…….
A: We have implemented the changes as recommended by the reviewer.
- Please read the full text and modify some minor issues, such as 1) Line 224, A a suspension of …; 2) Line 181 and 226 8 500 xg (There is space between 8 and 5), Line 202 Caffeine (1000 mg /ml, no space between 1 and 0) and Line 159 1,500 bp (a , between 1 and 5); 3) ml (Line 202, mg /ml) or mL(line 197, mg/mL), Please make sure.
A: We have implemented the changes as recommended by the reviewer.

Reviewer 3 Report
I think although the author put forward most of the problems, but there are still some problems need to be further modified.
1. In the Materials and Methods, “samples were pressurized at 100 and 200 400 MPa for 2 and 4 min”, please check whether the description of the processing conditions is correct.
2. The high hydrostatic pressure described in the figure notes is 100 MPa/4 min and 200 MPa/4 min and in the materials and methods are 100 and 200 400 MPa, but the abscissa in Fig.5B is 150 Mpa and 250 Mpa. What are the treatment conditions in this study?
3. There should be a space between the number and the unit, for example, Line 125, “10mM”, Line 214, “2.4mg/L” ...
4. “μl” should be “μL”, for example, Line 221, 230...
5. Line 241, “Fig 1” should be “Fig. 1”, there are many similar problems in the manuscript, for example, Line 372, Fig 3, Line 372, (Fig 4A), Line 609, (Fig 8A and B).
I think although the author put forward most of the problems, but there are still some problems need to be further modified.
1. In the Materials and Methods, “samples were pressurized at 100 and 200 400 MPa for 2 and 4 min”, please check whether the description of the processing conditions is correct.
2. The high hydrostatic pressure described in the figure notes is 100 MPa/4 min and 200 MPa/4 min and in the materials and methods are 100 and 200 400 MPa, but the abscissa in Fig.5B is 150 Mpa and 250 Mpa. What are the treatment conditions in this study?
3. There should be a space between the number and the unit, for example, Line 125, “10mM”, Line 214, “2.4mg/L” ...
4. “μl” should be “μL”, for example, Line 221, 230...
5. Line 241, “Fig 1” should be “Fig. 1”, there are many similar problems in the manuscript, for example, Line 372, Fig 3, Line 372, (Fig 4A), Line 609, (Fig 8A and B).
Author Response
I think although the author put forward most of the problems, but there are still some problems need to be further modified.
- In the Materials and Methods, “samples were pressurized at 100 and 200 400 MPa for 2 and 4 min”, please check whether the description of the processing conditions is correct.
A: We sincerely appreciate the reviewer's comments regarding our study. The final pressures applied during our investigation were 150, 250, and 400 MPa. We observed no significant differences at the 2-minute mark for both 150 and 250 MPa. However, at the 4-minute interval, we did identify significant differences. It is important to note that no growth was observed at either the 2 or 4-minute marks when subjected to 400 MPa.
- The high hydrostatic pressure described in the figure notes is 100 MPa/4 min and 200 MPa/4 min and in the materials and methods are 100 and 200 400 MPa, but the abscissa in Fig.5B is 150 Mpa and 250 Mpa. What are the treatment conditions in this study?
A: We modified the material and methods to 150 and 250 MPa into manuscripts.
- There should be a space between the number and the unit, for example, Line 125, “10mM”, Line 214, “2.4mg/L” ...
A: A space has been incorporated into manuscripts (Line 126 and 215).
- “μl” should be “μL”, for example, Line 221, 230...
A: The μl has been change to “μL” into manuscripts (Line 222, 231).
- Line 241, “Fig 1” should be “Fig. 1”, there are many similar problems in the manuscript, for example, Line 372, Fig 3, Line 372, (Fig 4A), Line 609, (Fig 8A and B).
A: It has been incorporated a dot in the Fig.1, Fig.3, Fig.4 and Fig.8.
